# Optics and Utility of Low-Cost Smartphone-Based Portable Digital Fundus Camera System for Screening of Retinal Diseases

**DOI:** 10.3390/diagnostics12061499

**Published:** 2022-06-20

**Authors:** K. V. Chalam, Joud Chamchikh, Suzie Gasparian

**Affiliations:** Department of Ophthalmology, Loma Linda University School of Medicine, Loma Linda, CA 92354, USA; jchamchick@ucla.edu (J.C.); suzie.kazaryan@gmail.com (S.G.)

**Keywords:** optical principles, portable fundus camera, phone-based fundus camera, smartphone

## Abstract

Purpose: To describe optical principles and utility of inexpensive, portable, non-contact digital smartphone-based camera for the acquisition of fundus photographs for the evaluation of retinal disorders. Methods: The digital camera has a high-quality glass 25 D condensing lens attached to a 21.4-megapixel smartphone camera. The white-emitting LED light of the smartphone at low illumination levels is used to visualize the fundus and limit source reflection. The camera captures a high-definition fundus (5344 × 4016) image on a complementary metal oxide semiconductor (CMO) with an area of 6.3 mm × 4.5 mm. The auto-acquisition mode of the device facilitates the quick capture of the image from continuous video streaming in a fraction of a second. Results: This new smartphone-based camera provides high-resolution digital images of the retina (50° telescopic view) in patients at a fraction of the cost (USD 1000) of established, non-transportable, office-based fundus photography systems. Conclusions: The portable user-friendly smartphone-based digital camera is a useful alternative for the acquisition of fundus photographs and provides a tool for screening retinal diseases in various clinical settings such as primary care clinics or emergency rooms. The ease of acquisition of photographs from a continuously streaming video of fundus obviates the need for a skilled photographer.

## 1. Introduction

Conventional fundus photography assesses the quality of the media, optic nerve, retinal vasculature, and choroid and is commonly used in clinical practice for the diagnosis and monitoring of retinal diseases [1]. Traditionally, acquisition of high-resolution images is dependent on a high-quality fundus camera (typically an expensive, non-portable desktop system) and an experienced photographer who is familiar with both the imaging system and retinal anatomy.

More recently, with technological advances, handheld fundus cameras are gaining popularity as they are cost-effective, portable, and easy-to-handle with an improving quality of retinal photographs [2]. Validated handheld portable cameras currently available include the Volk Pictor Plus (Volk Optical, Mentor, OH, USA), Zeiss Visucoat 100 (Carl Zeiss, Jena, Germany), and Smartscope Pro (Optomed, Oulu, Finland) [3,4,5]. The images captured with these portable devices can be directly sent to ophthalmologists. Screening in such a manner would allow for the detection of early signs of retinal disease and timely medical intervention and prevention of visual impairment in sight-threatening diseases [6].

Smartphone-based fundus cameras are a subset of portable cameras and include the Volk iNview and Vistaview (Volk Optical, Mentor, OH, USA), Remidio Fundus on Phone (FOP, Remidio Innovative Solutions, Bangalore, India), and Make in India Retinal Camera (MII Retcam) [2,7,8]. Since many patients are routinely followed in primary care settings, a cost-effective and practical tool such as a smartphone-based digital camera is necessary in order to successfully screen for retinal diseases such as diabetic retinopathy [6,9]. However, the underlying optical principles of these instruments or limitations in quality of images from associated aberrations are not described.

In this report, we describe a novel, low-cost, easy-to-use, portable, noncontact digital fundus smartphone-based camera for screening of retinal disorders in specialty clinic and ambulatory (bedside or primary care, emergency room) settings. We additionally illustrate the optics principles involved in the production and dissemination of the retinal images captured with a handheld imaging system along with associated limitations.

## 2. Methods

### 2.1. Optics and Design

#### Optics of Retinal Image Capture

The optics of fundus imaging using a smartphone camera is similar in principle to retinal image acquisition with an indirect ophthalmoscope. A beam of light is directed toward the patient’s retina, and the reflected rays off the retina are condensed to an inverted real aerial image with a handheld +15–+30 D lens. The examiner sees the real, inverted aerial image that is located 2–4 cm in front of the handheld lens, depending on the power of the lens (Figure 1).

Conventional fundus cameras are also designed based on the same optical principles; the camera’s film or digital sensor array is positioned to capture the aerial image. In our system, the camera’s flashlight replaces the indirect ophthalmoscope light source, and the camera recording function of the smartphone captures aerial images (substitutes for observer’s eye).

### 2.2. Description of Housing Hardware

Similar to an indirect ophthalmoscope, the ophthalmic device has two separate components. A lightweight metal tube with a 25D condensing lens is attached to a smartphone (Figure 2).

As with indirect ophthalmoscopy, the image is created in between the examiner, and the condensing funduscopic lens. In this case, the image is created 6.6 cm in between the 25D lens and the camera’s aperture. The 18 cm distance between the camera’s aperture and the 25D lens allows for a working distance of 50 mm from the subject’s pupil. The system allows −20D to +20D autofocus range of accommodation. The attachment allows the generation of a 50° fundus image assisted by a high-quality glass funduscopic 25D lens. The combination of white LED light and the mobile application-powered smartphone camera assist ophthalmoscope acquired fundus photographs. The white LED light of the smartphone serves as light source (5 to 10 lumens) and permits the smartphone to obtain wide-angle fundus photographs at extremely low light levels.

### 2.3. Optics of Image Capture with Smartphone Lens

The lens complex in the smartphone contains six elements with power of +198D. Cyclooefin Copolymer (COC) and OKP4 (used as lens material) are selected for structural stability, thermal stability, and limited moisture absorption with refractive index of 1.53 and 1.61, respectively. The second lens (negative) is made of OKP4 and neutralizes the spherical and chromatic aberration produced by strong convex lens. The fourth lens does the same. Off axis aberrations are limited by the inflection point of the sixth lens component.

The smart phone lens complex has 6.464 mm total track of lens, 5.067 mm effective focal length, and 0.279 mm back focal length, 2.8 f-number, and 76° field of view. The chief ray angle (CRA) is limited to 34°. The image height of 3.96 mm [larger than size of the image sensor (3.96 mm)] prevent the dark angle caused by the deviation of the image sensor off optical axis. The F number is set at 2.8 to allow half field of view of 36°. Aperture is placed at the entrance pupil to limit aberrations. The chief ray angle is limited to 34°, and the placement of lens complex along with IR filter limits aberrations (Figure 3).

The optical image stabilization built into the camera decreases unwanted hand movement and associated distortions. This enables photographs to be of high resolution while maintaining appropriate detail. DOF (distance between the nearest and farthest elements in a scene) is a function of the focal length, illumination and aperture. The aperture of the camera (range of ƒ/1.8–ƒ/22) permits fast shutter speed and shorter reboot time in between images. Higher depth of focus from small aperture allows the capture of pathology in the anterior vitreous without compromising the detail of the posterior pole. The smaller aperture size limits chromatic aberration by avoiding light dispersion.

### 2.4. Optics of Image Processing in Smartphone

Unlike traditional cameras, optical design of mobile phones is limited by thickness. Complementary metal oxide semiconductor (CMOS) is used to capture the image signal. Photodetectors coupled with amplifier capture the light signal. Analog signal is converted to digital signal instantaneously on the same chip with minimal power consumption and heat generation (unlike CCDs). CMOs is placed approximately 5 mm behind the lens complex. Pixel size (1.12 micrometer) and 21.4 mega pixel chip (image area of 6.3 × 4.5 mm) allows the high-definition capture of the image (5344 × 4016) at very low illumination levels. Since image area is 6.3 mm × 4.5 mm, image height is set at 3.96 mm.

For the pixel size of 1.12 um, the maximum spatial resolution is approximately 446.4 lp/mm. To eliminate optical aberrations, pixel size approximates Airy disk size (4.08 µm). Optical distortion is limited to less than 1%. With use of image space telecentric system, image magnification is constant, independent of the position of fundus image generated by 25D lens.

### 2.5. Image Processing Hardware

#### 2.5.1. Storage Capability and Photo Manipulation

The mobile application on the smartphone allows for the storage of the images on the smartphone; built-in software enables the optimization of the image with adjustments of luminance, color saturation, and hue balance. Red-free photograph mode is advantageous for detection of early-onset diabetic retinopathy [9]. Touch based finger manipulation zooms and magnifies fundus image. Each image creates a 100 Kbyte compressed JPEG file.

#### 2.5.2. Photo Transferability

The use of a mobile application powered smartphone is advantageous in that it allows the easy transfer of the image by directly connecting the smartphone to a computer or wirelessly via email or cross-platform messaging applications such as iMessage or WhatsApp. This facilitates infield work and long-distance transfer of high-resolution fundus pictures from remote areas.

## 3. Results

### Camera Operation

The structure of the smartphone system makes it easy to use by an inexperienced photographer. A button on the side of the device initiates smartphone-based fundus camera. After entry of patient demographics, camera’s viewfinder along with light is activated. The patient is comfortably seated in an examination chair. The user places the large end of the lens tube (fashioned like an indirect BIO lens) on the patient’s forehead. This allows the user to capture the image of the fundus and a clear view of the retina is obtained secondary to fixed distance of the tube at 160 mm (Figure 4). The software automatically captures the images in rapid succession.

During operation, the live video display on the smartphone screen helps the user align the device with the patient’s eye and steadily move the device towards the patient using red reflex from patient eye as guide. At the optimal position, the red reflex will quickly expand from a slit to fill the entire field-of-view. The software application analyzes the video stream throughout the entire time of operation and presents the photographs in the standard anatomical position for interpretation and localization of the pathology. When a focused retina is detected, retinal images are automatically acquired with flash photography and stored on the smartphone. The images stored on the memory card are transmitted to other computers via a USB cable or wireless connection. The auto-acquisition property of the smart phone system allows the quick capture of high-quality fundus pictures in a fraction of a second and obviates the need for a skilled photographer.

All research was performed in accordance with the principles stated in the Declaration of Helsinki. An untrained medical professional obtained fundus photos of both dilated eyes of all patients in the retina clinic. The photo acquisition took about 10 s per eye and image quality was evaluated based on a clear full 50-degree telescopic view of the retina. After completion of this, we obtained fundus photographs with traditional non-portable fundus camera. Both sets of photos were evaluated by the retina specialist. We successfully acquired over 1000 images with this instrument.

Images compared well with those acquired simultaneously with traditional fundus camera (Optos, Marlborough, MA, USA). We report eight representative images (Figure 5) captured in various clinic settings.

The quality of images (resolution, field) acquired with the smartphone system recorded in clinic and emergency room consultation settings are comparable to photographs acquired with a traditional fundus camera in normal as well as pathological conditions. Diagnosis of diabetic retinopathy or central retinal artery occlusion was made with a high degree of certainty from high-resolution fundus photographs obtained with the smart phone system

## 4. Discussion

We describe the optical principles and utility of a novel, inexpensive, portable, non-contact digital smartphone-based fundus camera for screening and evaluation of retinal pathology. This novel smart phone-based camera provides high-resolution digital images of the retina at a fraction of the cost of established, non-transportable, office-based fundus photography systems. It serves as a useful alternative for the acquisition of fundus photographs, providing a new means for screening of retinal diseases in a multitude of clinical settings.

Photographic screening protocols are used in primary and sub-specialty settings to diagnose and manage various retinal disorders and most often comprise the use of conventional fundus photography. This usually necessitates the acquisition of high-resolution images by a bulky, expensive, non-portable desktop system along with a trained photographer who is familiar with the use of the imaging system. The benefit of the classic conventional fundus camera includes obtaining photos of a wider region of the retina (up to 200°), allowing for the simultaneous evaluation of the central and peripheral retina. However, the use of a tabletop imaging system is not practical, particularly in emergency room settings and for the screening of retinal diseases overseas.

The utility of transportable fundus cameras is emerging as they are cost-effective, portable, and easy-to-use. Currently available examples of smartphone-based fundus cameras include the Volk iNview and Vistaview (Volk Optical, Mentor, OH, USA), Remidio Fundus on Phone (FOP, Remidio Innovative Solutions, Bangalore, India), and Make in India Retinal Camera (MII Retcam) [2,7,8]. In previous reports, fundus imaging using mobile and indirect ophthalmoscopy lenses have been described [9,10,11]. However, results have shown a limited resolution of fundus photos with variable sensitivity in screening retinal diseases. Additionally, the majority of the handheld devices have a limited field of view (up to 55°) [2].

Based on a smartphone-based mobile technology platform, we describe the utility of a modified smartphone camera that offers a portable means to capture quality fundus photographs easily and inexpensively. We obtained high-quality digital images of the retina up to 50° from the disc without contact with the subject’s eye. Moreover, an evenly illuminated field of view via the light allowed for fine retinal details to be seen and permitted the screening of posterior segment conditions. A continuous video stream allows the inexperienced personnel to obtain the photograph and obviates the need for skilled photographer. The low cost (USD 1000) allows for a wider access and promotes compliance in various clinical settings including the emergency rooms and primary care clinics. The lack of a stand or traditional table-mounted fundus camera offers the advantage of portability. For instance, it may be useful for ophthalmology residents on the inpatient consult service as moving ICU or bedbound patients to a stationary fundus camera are not practical. Many facilities additionally do not have easy access to such instruments and patients would need to be transported to the ophthalmology clinics, which could be difficult or impossible in certain scenarios. Digital images can be enhanced and shared quickly via mobile or wireless networks, and potentially could automatically be interpreted and sent for teleretinal screening. This could be an especially valuable tool for those practicing global medicine. We successfully evaluated over 1000 eyes with this instrument; images compared well to those acquired with the traditional fundus photography system (Figure 5).

In summary, our smartphone-based digital camera system is a useful, inexpensive, alternative tool to acquire fundus images. We outline the underlying optical principles involved in the generation of images and describe electronic technology used in storage and dissemination with the use of smart phone-based technology. It constitutes a cost-effective way for regular specialty clinic and ambulatory clinics to screen for diabetic retinopathy and other fundus pathologies in at-risk populations in lower income communities globally.

## Figures and Tables

**Figure 1 diagnostics-12-01499-f001:**
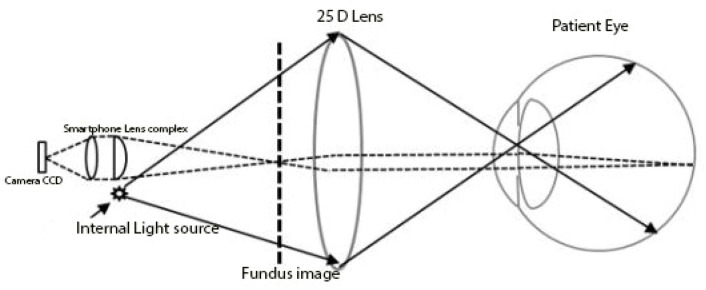
Schematic Drawing Illustrating the Viewing System of the smartphone Retinal Imager. The digital camera has a 25D condensing lens attached to a 21.4-megapixel camera with a built-in lens complex. The smart phone camera’s flash provides the internal light source with a real image created between the 25D lens and smartphone.

**Figure 2 diagnostics-12-01499-f002:**
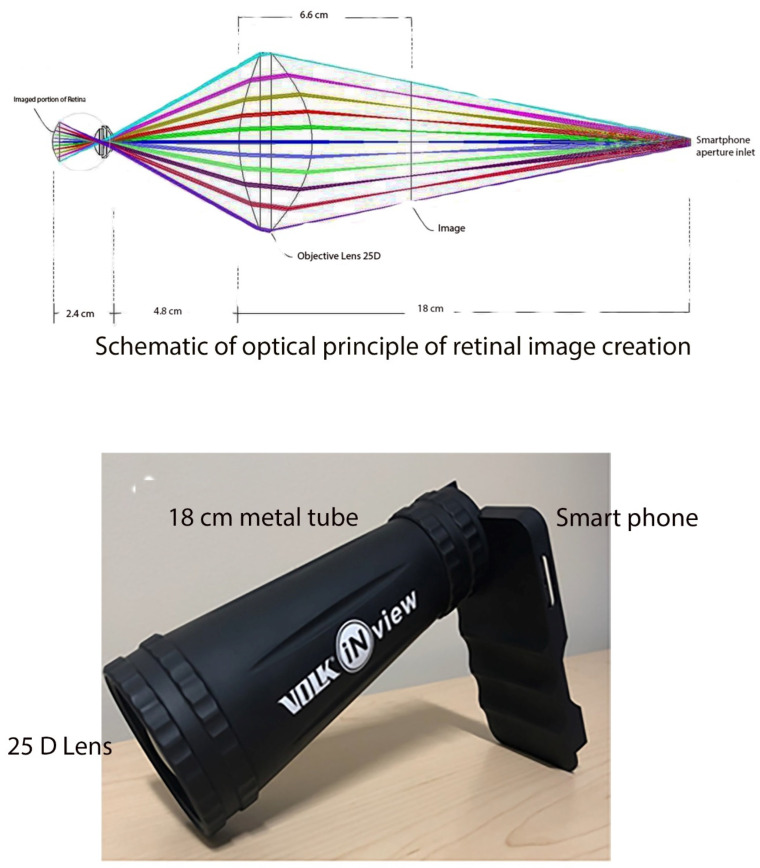
Housing and optical principles of smartphone retinal imager. The 25D condensing lens sits approximately 4.8 cm from the eye at the farthest end of the 18 cm metal tube, which is attached to the smartphone. The smartphone-based fundus camera allows for creation of an inverted, real aerial image approximately 6.6 cm from the condensing lens.

**Figure 3 diagnostics-12-01499-f003:**
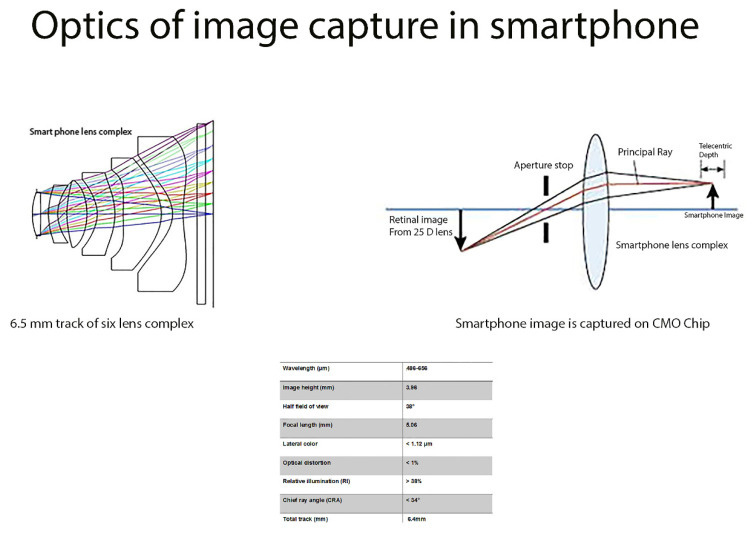
The smart phone lens complex has approximately 6.5 mm total track of lens with about 5.067 mm effective focal length and image height of 3.96 mm. The smart phone image is captured on the CMO chip.

**Figure 4 diagnostics-12-01499-f004:**
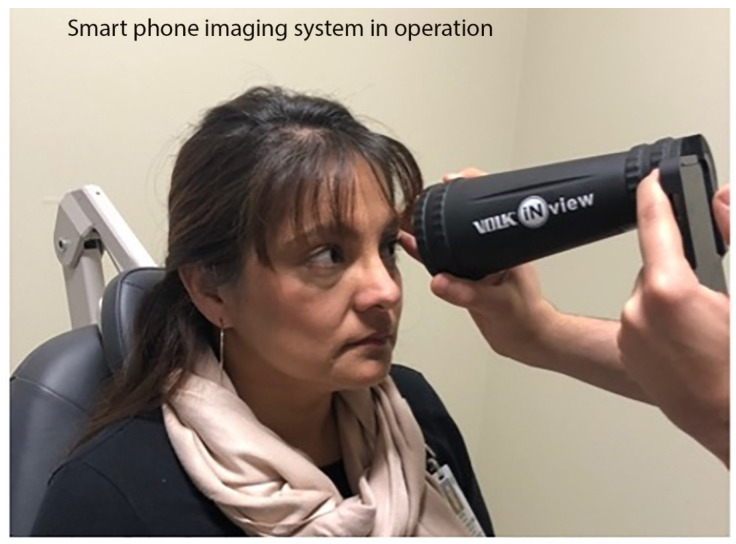
Smartphone Retinal Imager in Operation. Volk iNview Retinal Imager in use, stabilized on a patient’s forehead owing to the BIO lens-like end tube structure allowing for fundus photography to be obtained in any clinical setting without need of a slit lamp or specialized equipment.

**Figure 5 diagnostics-12-01499-f005:**
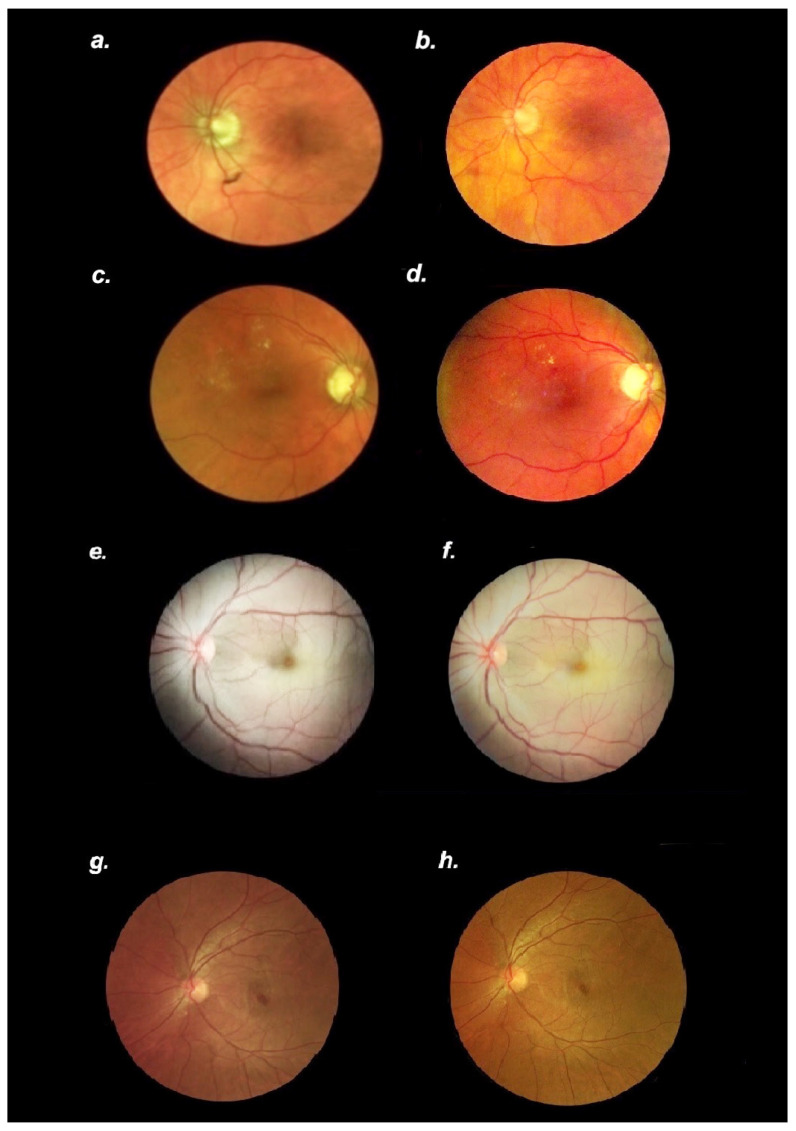
Side-by-side Comparison of Fundus Photography Utilizing the traditional fundus camera (Optos California) and Smartphone Retinal Imager. (**a**). Normal fundus photograph of a left eye using the Optos fundus camera (**b**). Normal fundus photograph of the same left eye using the smartphone retinal Imager (**c**). Fundus photograph of the right eye demonstrating diabetic retinopathy with macular edema taken using the Optos fundus camera (**d**). Fundus photograph of the same right eye demonstrating diabetic retinopathy with macular edema taken with the smartphone retinal Imager (**e**). Fundus photograph of the left eye demonstrating central retinal artery occlusion taken using the Optos fundus camera (**f**). Fundus photograph of the same left eye demonstrating central retinal artery occlusion taken with the smartphone retinal Imager (**g**). Fundus photograph of the left eye demonstrating bull’s eye maculopathy secondary to hydroxychloroquine use taken using the Optos fundus camera (**h**). Fundus photograph of the same left eye demonstrating bull’s eye maculopathy secondary to hydroxychloroquine use taken with the smartphone retinal Imager.

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
