# Peer review of "Optics and Utility of Low-Cost Smartphone-Based Portable Digital Fundus Camera System for Screening of Retinal Diseases"

_diagnostics, 2022, doi:10.3390/diagnostics12061499_

Round 1
Reviewer 1 Report
The topic is current. Smartphone based digital camera system is a useful, portable and low-expensive tool to acquire fundus images. It is handheld but not comfortable for sure. Anyway, the quality as well as the resolution of acquired images are very good. In the current post-Covid new world, this system might be very useful for regular specialty clinic and ambulatory clinics, especially in poor countries. The manuscript is well-written and well-organized. The reference list is updated.
Author Response
Thank you for your kind comments
Reviewer 2 Report
The authors describe the optics (very detailed; optics are similar to indirect ophthalmoscopy) and the utility (appr. 1,000 images of over 1,000 eyes; photo acquisition is 10 seconds per eye) of a low cost (US$ 1,000) smartphone-based portable digital fundus camera system for screening of retinal diseases. This device seems to be very feasible for primary care settings or emergency rooms. Photos or videos might be submitted easily to ophthalmic professionals.
All images compared well with those taken simultaneously with a traditional fundus camera (Fig.5. shows 4 comparisons; Photos from the smartphone retinal imager don´t look inferior compared with those of the Optos fundus camera, even sharper/better; easy to interpret the retinal diseases).
In summary, a well written and interesting to read manuscript with clinical relevance.
Just 1 comment: maybe the authors could add the pixel size and the number of mega pixel of the Optos camera used for the photos in Fig.5. [Smartphone: Pixel size (1.12 micrometer) and 21.4 mega pixel chip].
Author Response
Thank you for your kind comments. Optos is a confocal system with resolution of 20 microns (equal to 12 mega pixel camera of a flash based system)